# Conidium Specific Polysaccharides in *Aspergillus fumigatus*

**DOI:** 10.3390/jof9020155

**Published:** 2023-01-24

**Authors:** Zhonghua Liu, Isabel Valsecchi, Rémy A. Le Meur, Catherine Simenel, J. Iñaki Guijarro, Catherine Comte, Laetitia Muszkieta, Isabelle Mouyna, Bernard Henrissat, Vishukumar Aimanianda, Jean-Paul Latgé, Thierry Fontaine

**Affiliations:** 1Institut Pasteur, Unité des Aspergillus, 75015 Paris, France; 2DYNAMYC 7380, Faculté de Santé, Université Paris-Est Créteil (UPEC), 94010 Créteil, France; 3Institut Pasteur, Université Paris Cité, Centre National de la Recherche Scientifique (CNRS) UMR3528, Biological NMR and HDX-MS Technological Platform, 75015 Paris, France; 4Institut Pasteur, Université Paris Cité, Unité de Biologie des ARN des Pathogènes Fongiques, 75015 Paris, France; 5Architecture et Fonction des Macromolécules Biologiques, CNRS, Aix-Marseille Université Marseille, 163 Avenue de Luminy, CEDEX 09, 13288 Marseille, France; 6Institut Pasteur, Université Paris Cité, CNRS UMR2000, Unité de Mycologie Moléculaire, 75015 Paris, France; 7Institut Pasteur, Université Paris Cité, INRAE, USC2019, Unité Biologie et Pathogénicité Fongiques, 75015 Paris, France

**Keywords:** *Aspergillus fumigatus*, cell wall, conidium, polysaccharidome

## Abstract

Earlier studies have shown that the outer layers of the conidial and mycelial cell walls of *Aspergillus fumigatus* are different. In this work, we analyzed the polysaccharidome of the resting conidial cell wall and observed major differences within the mycelium cell wall. Mainly, the conidia cell wall was characterized by (i) a smaller amount of α-(1,3)-glucan and chitin; (ii) a larger amount of β-(1,3)-glucan, which was divided into alkali-insoluble and water-soluble fractions, and (iii) the existence of a specific mannan with side chains containing galactopyranose, glucose, and N-acetylglucosamine residues. An analysis of *A. fumigatus* cell wall gene mutants suggested that members of the fungal GH-72 transglycosylase family play a crucial role in the conidia cell wall β-(1,3)-glucan organization and that α-(1,6)-mannosyltransferases of GT-32 and GT-62 families are essential to the polymerization of the conidium-associated cell wall mannan. This specific mannan and the well-known galactomannan follow two independent biosynthetic pathways.

## 1. Introduction

The human fungal pathogen *Aspergillus fumigatus* produces a high number of asexual spores (conidia) that are dispersed into the air. The daily inhalation of these conidia leads to a wide spectrum of diseases, from simple rhinitis in immunocompetent hosts to fatal invasive aspergillosis in immunocompromised individuals [1,2]. During the infectious cycle, inhaled conidia initiate the germination process to form hyphae. *A. fumigatus* cell wall is the first fungal component to interact with the host tissue, modulate the host immune response, and protect fungal cells against host defenses. *A. fumigatus* cell wall studies have been almost exclusively focused on the mycelium, while the infective propagule that trigger the early immune response towards *A. fumigatus* is the conidium. It is mainly known that the outer layers of the conidium and mycelium are very different. In contrast to hyphae, conidia are covered by a melanin-pigment layer and a protein rodlet surface layer which confer immunological inertia toward the host immune system [3,4,5,6,7,8]. The underlying conidial polysaccharides have been very poorly investigated. Early studies have shown that the polysaccharide cell wall composition of the conidium and mycelium is different. For example, galactosaminogalactan is absent from resting conidia and starts being produced during the germination process [9]. The conidial cell wall contains a low amount of the polysaccharides chitin and α-(1,3)-glucan [10]. However, a comprehensive study of the conidial polysaccharides has not yet been undertaken. 

Our recent analysis of the α-mannosyltransferases in *A. fumigatus* has suggested that the cell wall of the resting conidia possesses a specific cell wall mannan [11,12]. In the present study, therefore, we investigated the polysaccharidome of the conidia cell wall using biochemical and biophysical approaches and showed how it was differently organized than the mycelial cell wall. The conidial cell wall displays a specific polymer fractionation, with an alkali-insoluble, as well as a water-soluble fraction that both contain mainly β-(1,3)-glucan and mannans. Moreover, we identified a new mannan in the water-soluble fraction, which contains side chains composed of galactopyranose, glucose, and N-acetylglucosamine residues. The characterization of the latter water-soluble fraction allowed us to identify a new conidia-specific mannan structure. 

## 2. Materials and Methods

### 2.1. Fungal Strains and Growth Media

The *A. fumigatus* strains used in this study are listed in Appendix A. The *A. fumigatus* parental strain KU80Δ*pyrG* (auxotrophic to uridine and uracil) was generated from the clinical isolate CBS 144-89 [13]. Conidia were harvested from cultures on 2% malt agar slants maintained at room temperature for 14 days or from *Aspergillus* minimal medium (AMM [14], 1% phytagel, Sigma, Saint-Louis, MS, USA) at room temperature for 4 weeks using 0.05% (*v*/*v*) Tween-20, and were filtered through a 40 μm nylon cell strainer to remove mycelial fragments (BD Falcon). Conidia grown on malt medium were used for the cell wall analysis, and conidia grown on the AMM/phytagel medium were used to isolate and characterize the G3Man polymer.

### 2.2. Fractionation of Cell Wall Polymers

Resting conidia were disrupted with 0.3 mm glass beads in a 0.2 M Tris-HCl buffer, pH 8.0, with a B. Braun cell disruptor (Melsungen GA, Hessen, Germany). A crude cell wall fraction was collected by centrifugation (4500× *g*, 10 min) and washed three times with distilled water. Cell wall proteins were extracted twice from the pellet by boiling in a 50 mM Tris-HCl, 50 mM EDTA buffer (pH 7.5) containing 2% SDS, and 40 mM β-mercaptoethanol for 15 min. After centrifugation (4500× *g*, 10 min) and three washes with 0.2 M NaCl, the cell wall pellet was incubated in 1 M NaOH containing 1 mg/mL NaBH_4_ at 65 °C for 1 h. After centrifugation at 4500× *g* for 10 min, the pellet was processed a second time under the same alkaline conditions, washed four times with distilled water, freeze-dried, and collected as the alkali-insoluble (AI) fraction. The alkali-soluble (AS) supernatant was neutralized by the addition of acetic acid and dialyzed extensively against water. The dialysate (AS fraction) was centrifuged (4500× *g*, 10 min), which resulted in two fractions: a water-soluble supernatant (ASSN) and a water-insoluble pellet (ASP). All fractions were freeze-dried and stored at room temperature.

### 2.3. Purification of Mannan from the ASSN Fraction of Conidia Cell Wall

The ASSN fraction (100 mg) was digested by a recombinant endo-β-1,3-glucanase [laminarinase A (LamA) from *Thermotoga neapolitana*, activity 10 µmol eq. glucose/min [15]] for 2 days in 50 mM sodium acetate (pH 6.0) and 10 mM sodium azide. The sample was dialyzed against water, freeze-dried, and submitted to an anion exchange chromatography on a Q-Sepharose column (Q-FF HiTrap, 5 mL, GE-Healthcare) equilibrated with 10 mM Tris-HCl (pH 7.5) at a flow rate of 1 mL/min. After the elution of the unbound fraction, the bound fraction was eluted with the following NaCl gradient: 0 to 0.25 M in 30 min, 0.25 M to 0.5 M in 10 min, and finally, 10 min under isocratic conditions with 0.5 M NaCl in a 10 mM Tris-HCl pH 7.5 buffer. The sugar-positive unbound fraction containing mannan was dialyzed and purified further by gel filtration on a Superdex 200 column (16 × 600 mm, GE Healthcare, Chicago, IL, Us) equilibrated in 150 mM ammonium acetate pH 4.0 at a flow rate of 0.5 mL/min. 

### 2.4. Carbohydrate Analysis of Cell Wall Fractions

To analyze the monosaccharide composition of different fractions from the resting conidial cell wall, the total neutral hexoses were quantified by the phenol-sulfuric procedure using glucose as a standard [16]. Neutral monosaccharides were analyzed by gas–liquid chromatography (GC) as alditol acetates obtained after hydrolysis (4 N trifluoroacetic acid, 100 °C, 4 h), reduction, and peracetylation [17]. Derivatized monosaccharides were separated and quantified on a DB5 capillary column (25 m × 0.32 mm, SGE) using a Perichrom GC apparatus (carrier gas, 0.7 bar helium; temperature program, 120–180 °C at 2 °C/min and 180–240 °C at 4 °C/min). To quantify amino-sugars, cell wall fractions were hydrolyzed with 6 N HCl at 100 °C for 6 h. After drying the hydrolysate under a vacuum, amino-sugars in the samples were analyzed by high-performance anion exchange chromatography (HPAEC) with a pulsed electrochemical detector and an anion exchange column (CarboPAC PA-1, 4.6 × 250 mm, Dionex) using 18 mM NaOH as the mobile phase at a flow rate of 1 mL/min; glucosamine and galactosamine were used as standards [11]. Glycosidic linkages were investigated by the methylation of cell wall fractions followed by GC-MS analysis as previously described [18,19].

To quantify β-(1,3)-glucan and α-(1,3)-glucan in the cell wall, the AI, ASP, and ASSN fractions from resting conidia were submitted to enzymatic digestions. Fractions (1 mg/mL) were incubated with recombinant β-(1,3)-glucanase [LamA from *T. neapolitana* [15]] and the mutanase from *Trichoderma harzianum* [20], respectively, in a 20 mM sodium acetate buffer (pH 5.5) at 37 °C for 24 h. The reducing sugar released after enzyme digestion was quantified by the 4-hydroxybenzhydrazide (PABA) assay [21]. To quantify β-(1,6)-branching in β-(1,3)-glucan and β-(1,3)/(1,4)-glucan, the enzyme digests of different cell wall fractions were subjected to HPAEC using a CarboPAC PA-1 column (4.6 × 250 mm, Dionex) at a flow rate of 1 mL/min; eluent A was 50 mM NaOH, and eluent B was 0.5 M NaOAc in 50 mM NaOH. The elution gradient was: 0–2 min, isocratic 98% A:2% B; 2–15 min, linear gradient from 98% A:2% B to 65% A:35% B; 15–35 min, linear gradient from 65% A:35% B to 30% A:70% B, followed by 100% B for 3 min. The amount of cell wall galactomannan was estimated from the mannose and galactose content of the AI fraction [11,19].

### 2.5. Nuclear Magnetic Resonance (NMR) Analysis

NMR experiments were recorded at 308 K either on an 800 MHz Avance NEO with an 18.8 Tesla magnetic field or on a 600 MHz Avance III HD spectrometer with a 14.1 Tesla magnetic field, both from Bruker (Billerica, MA, USA). Spectrometers were equipped with cryogenically cooled triple resonance ^1^H[^13^C/^15^N] probes. Spectra were recorded using TopSpin 4.0.7 on the 800 MHz spectrometer or Topspin 3.6.1 for the 600 MHz spectrometer (Bruker Biospin). ^1^H and ^13^C chemical shifts were referenced to external DSS (2,2-dimethyl-2-silapentane-5-sulfonate, sodium salt).

Polysaccharide samples (5 to 20 mg) were dissolved either in 200 µL or 300 µL of D_2_O (99.97% ^2^H atoms, Eurisotop, Saclay, France) and placed in 3 mm tubes (Norell HT, Sigma-Aldrich) or 4 mm Shigemi tubes, depending on the amount of the sample. Resonance assignment, glycosidic bonds identification, and J coupling measurements were achieved from homonuclear ^1^H-^1^H COSY [22] and natural abundance heteronuclear ^1^H-^13^C experiments: HSQC spectra recorded with or without decoupling [23], H2BC [24], HMBC [25], and HSQC-TOCSY [26] with a 200 ms mixing time. Monosaccharide residues were identified from the anomeric region; their anomeric configuration was established from the corresponding chemical shifts and the ^1^J_H1,C1_ coupling constant. ^3^J_H1,H2_ coupling constants were measured from ^1^H-^1^H COSY experiments obtained with a resolution of 1 Hz. Glycosidic bonds were identified using HMBC experiments. The proportion of the different monosaccharide residues was estimated from the integrals of the anomeric peaks on the ^1^H-^13^C-HSQC spectrum.

Stimulated echo diffusion experiments of the yet undescribed mannan (G3Man, see Results) were performed at 298°K with convection compensation and bipolar gradients [27]. These diffusion-ordered experiments (DOSY) were recorded with a diffusion delay of 140 ms; gradients were applied during 3.4 ms with 16 varying intensities and a total recycle delay of 3 s. Experiments were performed in triplicate. The diffusion coefficients were calculated with Topspin DOSY standard routines.

The apparent diffusion coefficient (*d*) of mannan for selected individual signals was obtained by fitting the integral (*I*) of selected signals to gaussian decays as a function of the gradient intensity (*G*):$$I=Io\;e^{-dG^{2}}$$
where *Io* represents the signal that is integral in the absence of gradients. Integral errors were estimated with five times the noise standard deviation. Fits were performed with Kaleidagraph™ 4.5 (Synergy software).

### 2.6. Statistical Analysis

The one-way repeated measure (RM) ANOVA, two-way ANOVA, or t-test (nonparametric) were performed with Prism-8 (GraphPad Software, Inc., La Jolla, CA, USA).

## 3. Results

### 3.1. Polysaccharidome of the Conidial Cell Wall Is Unique

We analyzed the three cell wall fractions extracted from *A. fumigatus* conidia grown on malt agar: the alkali-insoluble (AI) fraction and the alkali-soluble fraction that were further separated into water-soluble (ASSN) and water-insoluble (ASP) fractions. AI and ASP were the major polysaccharide fractions. The ASSN fraction of the conidial cell wall represented >25% of the total cell wall polysaccharides (Figure 1), whereas the ASSN from the mycelium cell wall accounted for less than 2.5% (Appendix A). The conidial cell wall was characterized by a chitin content of 10% (established by the GlcNAc content of the AI fraction), a high glucan contents (>50%, represented by glucose), the absence of galactosamine, and a high amount of mannan polymers (more than 20% of total cell wall polysaccharides) (Figure 1A). Since glucans were the major cell wall polysaccharides in conidia, we investigated their composition and partition: α-(1,3)-Glucan was mainly found in the ASP fraction, and β-(1,3)-glucan was detected in all fractions. In contrast to our early findings in the analysis of the mycelium cell wall [19,28], a high amount of water-soluble β-(1,3)-glucan was present in the ASSN fraction (Figure 1A, Appendix A). This soluble β-(1,3)-glucan was eluted at the void volume of the Superdex 200 column (Mr > 1000 kDa), suggesting a differential organization of β-(1,3)-glucan in the conidial cell wall relative to the mycelial cell wall.

### 3.2. GH72 Family Is Involved in the β-Glucan Organization of the Conidial Cell Wall

The differential repartition and water-solubility of the conidial β-(1,3)-glucan present in the conidial cell wall was analyzed in β-glucanases and transglycosylases cell wall mutants (Appendix A). The deletion of genes coding for endo-β-(1,3)-glucanases (GH-16 and GH-81), exo-β-(1,3)-glucanases (GH-55), or transglycosylases (GH-17) had no major impact on the repartition of β-(1,3)-glucan in the cell wall of the conidia in these mutants (Appendix A). The amount of water-soluble β-(1,3)-glucan remained at ~25% of the total cell wall β-(1,3)-glucan in these mutants. Only a small decrease in the alkali-soluble water-insoluble β-(1,3)-glucan (ASP fraction) was observed in the GH55 and EndoG/ExoG mutants (Appendix A). In contrast, the deletion of the genes coding for β-(1,3)-glucanosyltransferases of the GH72 family modified the repartition as well as the amount of β-(1,3)-glucan in the conidial cell wall, confirming their role in β-(1,3)-glucan remodeling. The double deletion of *GEL1* and *GEL2* led to a decrease in β-(1,3)-glucan within the ASSN fraction and an increase in β-(1,3)-glucan within the ASP fraction, showing a decrease in β-(1,3)-glucan water solubility within the conidial cell wall of this mutant (Figure 2). The estimation of β-(1,3)-glucan branching and β-(1,3)/(1,4)-glucan by LamA digestion and HPLC analysis showed that the β-(1,3/(1,4)-glucan content was reduced in the *Δgel1-2* mutant (Figure 2C). Moreover, the cell wall alterations in the mutant conidia were associated with a significant increase in the α-(1,3)-glucan within the ASP fraction (Figure 2B). These results confirm that the GH72 β-(1,3)-glucanosyltransglycosylases play an important role in the glucan organization of the mycelial as well as conidial cell walls.

### 3.3. A New Mannan (G3Man) Structure Isolated from the ASSN Fraction of Conidia

The ASSN fraction showed an unexpected composition with four monosaccharides: glucose, galactose, mannose, and N-acetylglucosamine. To analyze this ASSN fraction, it was first subjected to LamA digestion to remove the contaminating β-(1,3)-glucans from this fraction. The LamA-treated fraction was passed through an anion exchange chromatography column. The unbounded fraction was the major fraction and contained >60% of the mannan content of the ASSN fraction. This unbound fraction was passed through a Superdex S200 gel filtration column, which eluted as two broad peaks, FNR-A and FNR-B, with apparent average molecular weights of 90 kDa and 37 kDa, respectively (Figure 3).

Sugar analysis showed that these two fractions (FNR-A and FNR-B) were composed of mannose, glucose, galactose, and N-acetylglucosamine (GlcNAc) in comparable ratios (Figure 3B). Methylation of both FNR samples followed by the analysis of partially methylated and acetylated alditols led to the identification of six methyl-ethers: three terminal sugars (glucose, galactose, GlcNAc), two monosubstituted hexoses, namely 4-O-substituted galactose and 6-O-substituted mannose, and 4,6-di-O-substituted mannose (Table 1), showing that both fractions corresponded to the same polymer with different Mr. The major fraction FNR-B was then fully characterized by NMR. NMR ^1^H-^13^C-HSQC analysis in the anomeric region revealed that FNR-B contained six major monosaccharide residues accounting for ~96% of the total anomeric signals (Table 2). Their sequence and glycosidic linkages were determined from ^1^H-^13^C HSQC and ^1^H-^13^C-HMBC experiments (Figure 4). Two α-mannose residues (A and F) were identified: residue A was substituted in positions four and six, and residue F was only substituted at position six. The analysis of the inter-residue connectivity showed that C1/H1 from each mannose residue were linked to C6/H6 of mannose residues (Figure 4A), indicating that mannose residues formed a linear α-(1,6)-mannan chain with side chain ramifications on position four. The integration of the corresponding anomeric signals showed that 92% of these mannoses (residue A) are substituted at position four, while the remaining 8% are not (residue F). Three monosaccharide residues were identified as linked to position four of the mannose residues: β-glucose (residue E) and β-galactose in a pyranose configuration (residues C and D) (Figure 4B). The α-GlcNAc residue (residue B) was not directly branched on the mannan chain but was linked to position four of the galactose residue C (Figure 4C). Altogether, NMR data were in agreement with the methylation analysis. This polymer is composed of a main α-(1,6)-mannan chain where 92% of mannose residues are 4-O-substituted (Figure 4D). More precisely, on the main chain, 52% of mannose residues were substituted with β-galactopyranose bearing itself a substitution at position four with an α-GlcNAc residue. In 24% of the mannose residues, the substitution occurred with a single β-galactose unit, while 16% of the mannose residues were substituted with a single β-glucose. No linkage between substituted (A) and unsubstituted (F) mannose residues was detected in ^1^H-^13^C HMBC spectra (Figure 4A), suggesting that the polymer contains a substitution-free (or several long) α-(1,6)-mannose sequence(s). The polysaccharide structure of G3M is summarized in Figure 5.

To assess whether the substitution-free and substituted α-(1,6)-mannose chains were part of the same polymer or not, we performed diffusion-ordered (DOSY) experiments by NMR. DOSY experiments can separate the spectra of molecules with a different diffusion coefficient and, hence, different hydrodynamic properties. Moreover, the width of signals on the diffusion pseudo-dimension (standard deviation) can reflect the size distribution of a polymer. We found that the polymer signals at every chemical shift had the same diffusion coefficient and standard deviation (94 ± 31 × 10^8^ cm^2^/s), indicating that the unsubstituted and substituted α-(1-6) mannose chains formed part of the same polymer (Figure 6). Most of the signals of residue F (unsubstituted α-(1,6)-mannose), which represents 8% of the total mannose residues, heavily overlapped with signals of the major form. However, the anomeric signal (4.96 ppm) only overlaps with the anomeric signal of residue E (β-glucose, 16% relative to mannose residues) and thus represents 33% of the signal. The diffusion decay of this signal can thus be used to compare the diffusion properties of the substituted/unsubstituted mannose chains, as shown in Figure 6B. The diffusion decay of F/E anomeric protons fits very well to a single gaussian decay (correlation coefficient R = 0.997) and is effectively the same as those of well-resolved signals of the substituted polymer, such as the anomeric ^1^H signal of A (4-substituted mannose at 5 ppm) and of the methyl group α-GlcNAc (B, 2.12 ppm), as shown by a 3.1% standard deviation from the mean for the three values. Therefore, the diffusion data strongly suggest that substituted and unsubstituted α-(1,6)-mannose residues belong to the same chain. Since the polymer contained glucosamine, galactose, and glucose linked to a mannan chain, we named it G3Man. Of note, we further verified that G3Man was absent from the mycelial cell wall.

The co-extraction of G3Man with β-(1,3)-glucan from conidia suggested their possible cross-linking, as shown for galactomannan and β-(1,3)-glucan co-isolation in the mycelium cell wall alkali-insoluble fraction [19,29]. To test the possible cross-linkage, the purification was performed without LamA digestion of β-(1,3)-glucan. The mannan polymer was eluted at the same retention time on the gel filtration Superdex 200 column, whereas β-(1,3)-glucan was excluded from the column (Appendix A). Moreover, an affinity chromatography on Concanavalin-A (specific binding of α-mannan) agarose matrix retained only the mannan polymer but not β-(1,3)-glucan (Appendix A), indicating that G3Man was not covalently linked to β-(1,3)-glucan or any other cell wall polysaccharide in the ASSN fraction. To verify if G3Man contains galactofuranose, we compared the G3Man produced by the *A. fumigatus* parental strain conidia and UDP-galactopyranose mutase deletion mutant (Δ*ugm1*) conidia [30]. The Δ*ugm1* mutant could not convert UDP-galactopyranose to UDP-galactofuranose and thus lacked galactofuranose. No structural difference was observed between the two G3Man, establishing the lack of galactofuranose in G3Man (Appendix A).

### 3.4. Biosynthesis of Conidial G3Man Is Dependent on OCH and MNN9/VAN1/ANP1

The presence of an α-(1,6)-mannan backbone in G3Man suggested that this chain could be synthesized by Och1-4 and Mnn9/Van1/Anp1 α-(1,6)-mannosyltransferases identified in *A. fumigatus* [12,31]. Accordingly, we showed that the quadruple Δ*och1-4* and the triple Δ*mnn9*/*van1*/*anp1* (ΔGT62) mutant conidia could not produce G3Man (Figure 7), demonstrating that both transferase families, OCH (GT-32) and GT-62, are essential for the mannan polymerization into G3Man.

## 4. Discussion

The polysaccharide composition of the cell wall of *A. fumigatus* conidia has been poorly investigated until this study. This lack of study is primarily due to the difficulty in collecting sufficient pure cell wall material free of pigments/proteins (present in high amounts in the conidial cell wall) to be able to undertake a comprehensive chemical analysis of the conidial cell wall. Previous studies have, however, shown that the cell wall outer layer of the conidia has a specific organization: the conidial surface is composed of melanin and proteinaceous rodlet layers that confer hydrophobicity to conidia and offer resistance to stress [3,5,6,7]. Conidial mannans are specifically important to control the conidial cell wall porosity and conidia survival [11,12,29].

Our polysaccharide analysis of the resting conidia cell wall has shown that the inner cell wall organization also differs between mycelia and conidia (Table 3). The lower amount of chitin and α-(1,3)-glucan of the conidia cell wall was compensated by a relatively high amount of β-(1,3)-glucan, galactomannan, and the presence of the soluble G3Man heteromannan. G3Man was absent from the mycelial cell wall. The structure of this polymer contained a linear chain of α-(1,6)-mannoside residues on which short side chains of α-N-acetylglucosamine-β-(1,4)-galactopyranose and β-galactopyranose or β-glucopyranose residue were attached. α-(1,6)-Mannan backbones have been described in a number of fungi [32,33,34] and have been well investigated in yeasts, where the main chain is substituted by side chains of one to six α-(1,2)-mannoside residues [35,36]. According to methylation and NMR data, the G3Man main chain is highly substituted, with more than 90% of 4-O-substituted mannose residues. The composition of the side chains of mannan polymers is quite variable and heterogenous in the fungal kingdom; however, to our knowledge, the substitution in position 4 on α-(1,6)-mannan main chains has never been reported.

Biosynthesis of the conidial G3Man was dependent on Ochp and Man9p/Van1p/Anp1p α-(1,6)-mannosyltransferases. In yeast, these glycosyltransferases are responsible for the N-mannan elongation of glycoproteins [37,38]. We did not find any role for these mannosyltransferases in the biosynthesis of mycelial mannan in a previous study [12]. However, it is clear from this work that these mannosyltransferases are essential for the construction of the conidia-specific mannan. Our data suggest that Ochp and GT-32 members are responsible for the polymerization of the α-(1,6)-mannan backbone of the G3Man and, therefore, suggest that G3Man is a part of N-glycan cell-wall glycoproteins. Unfortunately, we were unable to extract the cell wall G3Man without a destructive method (1M NaOH, 65 °C), preventing the identification of a G3Man-protein glycoconjugate but suggesting that G3Man was tightly attached to the cell wall AI core in its native form.

We searched for putative glycosyltransferases involved in the addition of mono-/disaccharide side chains on the conidial mannan backbone since none of the cell wall mutants in our hands had any modification of the G3Man structure. Unpublished in silico analysis of the *A. fumigatus* genome led to the identification of putative Golgi glycosyltransferases, which has not been characterized previously: AFUA_8G02690, AFUA_8G01730, and AFUA_5G09070 from the GT8 Cazy-family (http://www.cazy.org/ accessed on 15 November 2022) coding for putative α-N-acetyl-glucosaminyltransferases and AFUA_6G00520, AFUA_3G07220, and AFUA_2G17320 from the GT31 family, and coding for fungal-specific putative β-glycosyltransferases. All these six candidate genes are highly expressed in the conidia morphotype [39]. Single GT8 mutants and a triple GT31 mutant have been constructed. The fractionation of the cell wall of respective conidia and the purification of the FNR fraction from ASSN revealed that none of these genes are required for the side chain decoration of the G3Man mannan chain (Appendix A). The biological function of conidial G3Man remains unclear. Its biosynthetic defect has no effect on conidial production [12]. Its absence, however, leads to an increase in cell wall permeability and a decrease in conidia survival in vitro and in vivo [12]. In contrast to the mycelium cell wall, the conidium cell wall is characterized by a high amount of mannan. From our study, the conidial galactomannan (GM) and G3Man represent ~33% of the total cell wall polysaccharide (Figure 1 and Table 3) and follow two independent biosynthetic pathways.

The functions and activities of Golgi mannosyltransferases from GT-15 and GT-62 families have been investigated in *A. fumigatus*. GT-15 members are α-(1,2)-mannosyltransferases. Of the three members identified in *A. fumigatus*, CsmA/Ktr4p and CsmB/Ktr7p are essential for galactomannan biosynthesis [11,40]. The Ktr1/Mnt1p is involved in N- and O-glycans elongation [41,42] but not in galactomannan polymerization [11,42]. GT-62 members (Mnn9, Van1, Anp1) are α-(1,6)-mannosyltransferases. We showed that these activities are required for the polymerization of the G3Man conidia polymer but not for the cell wall galactomannan [12]. Recently, Oka and colleagues have shown that Anp1 (also called AnpA) is required for normal vegetative growth and the biosynthesis of the galactomannan [43], suggesting that mannan biosynthetic pathways could be more complex. As an example, galactomannan, produced by *A. fumigatus*, is a unique fungal polymer that is localized at the plasma membrane through a GPI anchor, at the cell wall covalently linked to the β-(1,3)-glucan network and is also released in the culture medium as a free form [19,44,45]. The deletion of CSMA/KTR4 or CSMB/KTR7 leads to the absence of galactomannan in the cell wall and in the medium [11,40]. However, a membrane-bound galactomannan was still identified in these mutants [46], but its structure was totally altered, indicating that other mannosyltransferase activities may use lipid acceptor to synthesize other mannan structures and compensate for the galactomannan defect [46]. Around 20 putative Golgi α-mannosyltransferases have been identified in the *A. fumigatus* genome. The comprehension of their functions in biosynthetic pathways of mannan structures in *A. fumigatus* requires further investigation.

Another major distinctive characteristic of cell wall polysaccharide organization found in conidia is the presence of β-(1,3)-glucan in two cell wall fractions: AI and ASSN. Since only one FKS1 glucan synthase gene is involved in β-(1,3)-glucan biosynthesis in *A. fumigatus*, the presence of water-soluble β-(1,3)-glucan in the ASSN fraction results from post-synthetic events. Many β-glucanases and transglycosylases are expressed in the conidia and are involved in the maturation of the conidial cell wall [39,47,48]. However, carbohydrate analyses of the conidia of the corresponding mutants showed that members of the GH17, GH16, GH81, and GH55 families do not have a major role in the cell wall organization of β-(1,3)-glucan. Although some members of the GH16, GH81, and GH55 families are involved in the pruning of β-(1,3)-glucan, which is required for chain separation during maturation in conidia (29, 38), these are not involved in the rearrangement of β-(1,3)-glucan in the inner layer of the cell wall. The only significant effect on β-(1,3)-glucan fractionation was observed in a GH72 deficient mutant (Figure 2). The GH72 family members are fungal transglycosylases that possess a dual activity and are able to elongate and branch into β-(1,3)-glucan chains [49]. In *A. fumigatus*, seven homologous genes have been identified, and only three (*GEL1*, *GEL2,* and *GEL4*) were expressed in laboratory conditions. These activities are essential in cell wall remodeling during vegetative growth [50]. The double *GEL1*/*GEL2* deletion led to an alteration of the conidial cell wall with a strong reduction in the melanin layer [51], a reduction in the ASSN β-(1,3)-glucan content, and a decrease in β-(1,3)/(1,4)-glucan of the conidia cell wall (Figure 2). The strong increase in the α-(1,3)-glucan content of the conidia cell wall of Δgel1/Δgel2 mutants (Figure 2) suggests a compensatory effect through the activation of the cell wall integrity pathway as has previously been observed in the *Aspergillus* species [52,53]. Moreover, *GEL4* is essential to *A. fumigatus* growth, suggesting that β-(1,3)-glucan remodeling is crucial for the cell-wall structuration [54]. All these chemical and enzymatic analyses of the cell wall mutants showed that the cell wall structure was permanently restructured during the life cycle. These continuous changes in the cell wall structure have been also confirmed by the analysis of the cell wall of swollen germinating conidia (early stages of germinating conidia without germ tubes). The composition of the swollen conidia showed a doubling of the cell wall polysaccharide content during swelling due to the chitin (+140%), β-1,3-glucan (+100%), and α-1,3-glucan (+500%) increase and the loss of G3Man and the initiation of the synthesis of GAG. Such changes in the composition of the cell wall can be also aligned with the up and down-regulation of cell wall enzymes as seen during morphogenetic changes [39,55].

Such differences can lead to differences in the immune response against the different fungal stages. To date, the immunomodulatory function of *A. fumigatus* cell wall polysaccharides has been almost exclusively investigated with polymers from the mycelium cell wall. The infective propagule initiates the disease in humans, whereas the infective propagule is the conidium. Our data showing that the composition of the conidial cell wall is different from the mycelial cell wall suggests that the immune reactions against the conidial cell wall polysaccharides may be specific and different from the ones induced by the mycelial cell wall. Preliminary analysis of the immune reactions against conidial polysaccharides suggest that this is the case. For example, G3Man did not induce any significative release of TNF-α; IL-6; IL-8, IL-1β, and IL-1Ra upon interaction with peripheral blood mononuclear cells even at a concentration as high as 50 µg/mL. The role of conidia-specific soluble polymers [β-(1,3)-glucans and G3Man] in the host immune response should be now investigated in depth and compared to the mycelial ones. Since conidial immune-masking rodlet and melanin layers are degraded [56] and new immunogenic polysaccharides [α-(1,3)-glucans and galactosaminogalactan (GAG)] are synthesized and exposed on the surface when the conidia germinate [9,20,57], the release in vivo of conidial soluble polymers must also be investigated to better understand the early host immune response against *A. fumigatus*.

## Figures and Tables

**Figure 1 jof-09-00155-f001:**
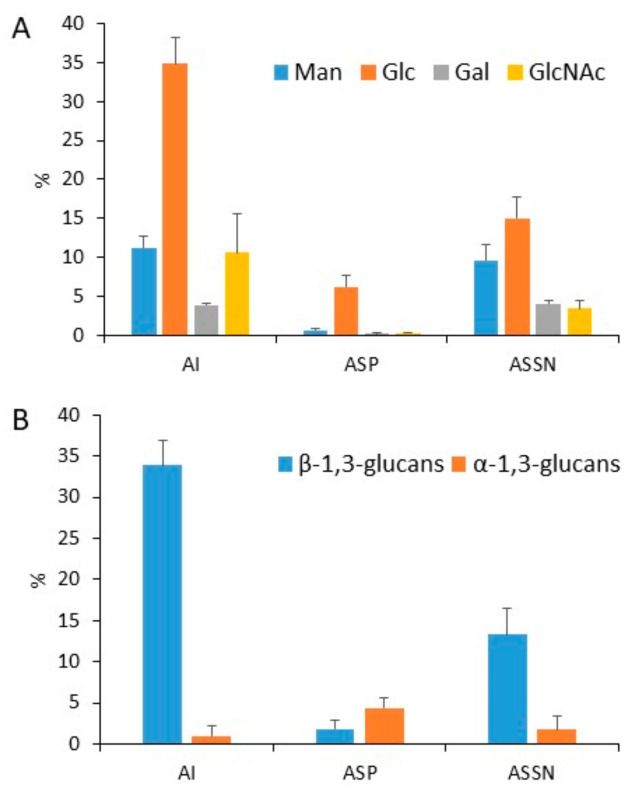
Monosaccharide (**A**) and Glucan (**B**) composition of the three cell wall fractions (AI, ASP, and ASSN) of conidia grown on malt medium; compositions are expressed as the percentage of the total cell wall sugar. The quantification of α- and β-(1,3)-glucans was performed by measuring the amount of reducing sugar released after endo-β-(1,3)-glucanase (LamA) and α-(1,3)-glucanase mutanase digestions using the PABA reagent. The experiments were performed on three independent biological replicates.

**Figure 2 jof-09-00155-f002:**
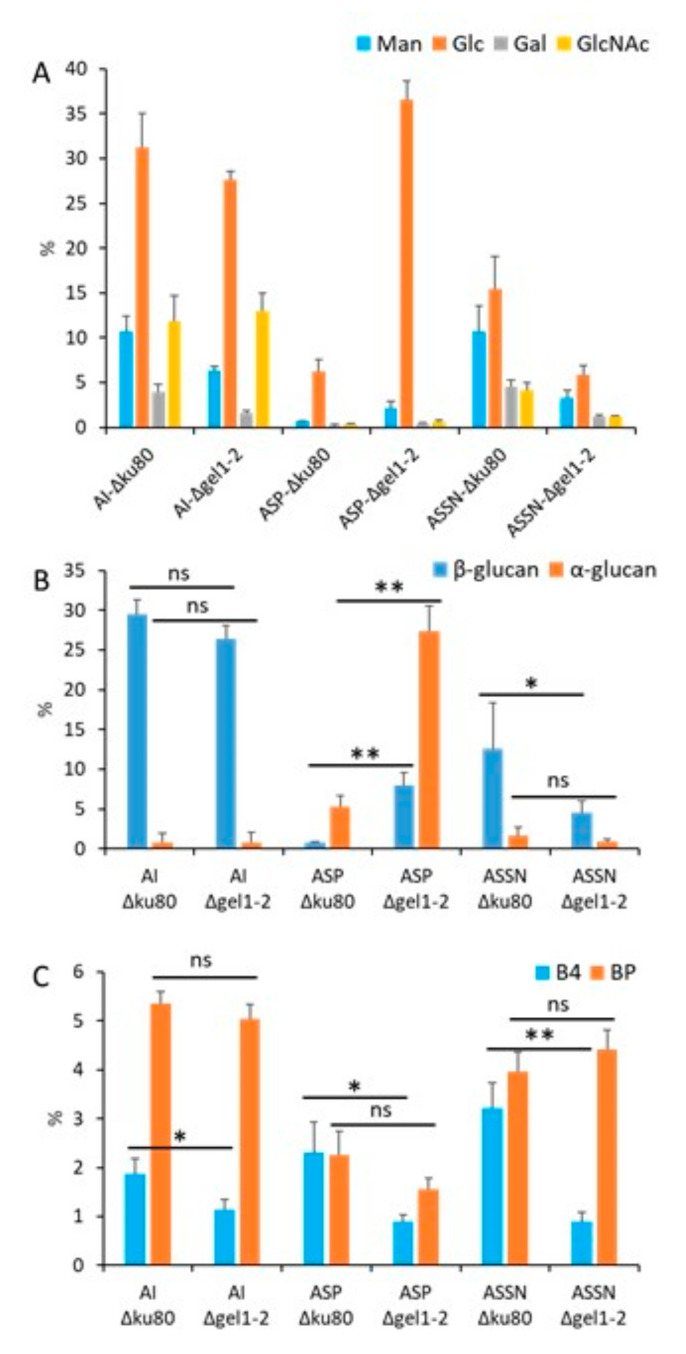
Sugar analysis of the conidial cell wall fractions of Δ*ku80* (parental strain) and Δ*gel1-2.* (**A**) Monosaccharide composition, (**B**) Distribution of the cell wall α- and β-glucans, and (**C**) Percentage of branching of β-(1,3)-glucans and β-(1,3)/(1,4)-glucan in the conidial cell wall fractions. (**A**,**B**) The percentage is relative to the total cell wall sugar. The quantification of α- and β-(1,3)-glucans was performed by measuring the amount of reducing sugar released after endo-β-(1,3)-glucanase (LamA) and mutanase digestions using the PABA reagent. The branching levels of β-(1,3)-glucan and β-(1,3)/(1,4)-glucan were estimated by measuring by HPLC the degradation products after LamA digestion (BP is the LamA-resistant branching point from the β-(1,6)-branched-β-(1,3)-glucan and BP4 is the LamA-resistant trisaccharide (β-Glc-(1,4)-β-Glc-(1,3)-Glc) from the β-(1,4)/(1,3)-glucan). The experiments were performed on three independent biological replicates. Statistical analysis by one repeated measure ANOVA: ns, non significative; * *p* < 0.033; ** *p* < 0.002.

**Figure 3 jof-09-00155-f003:**
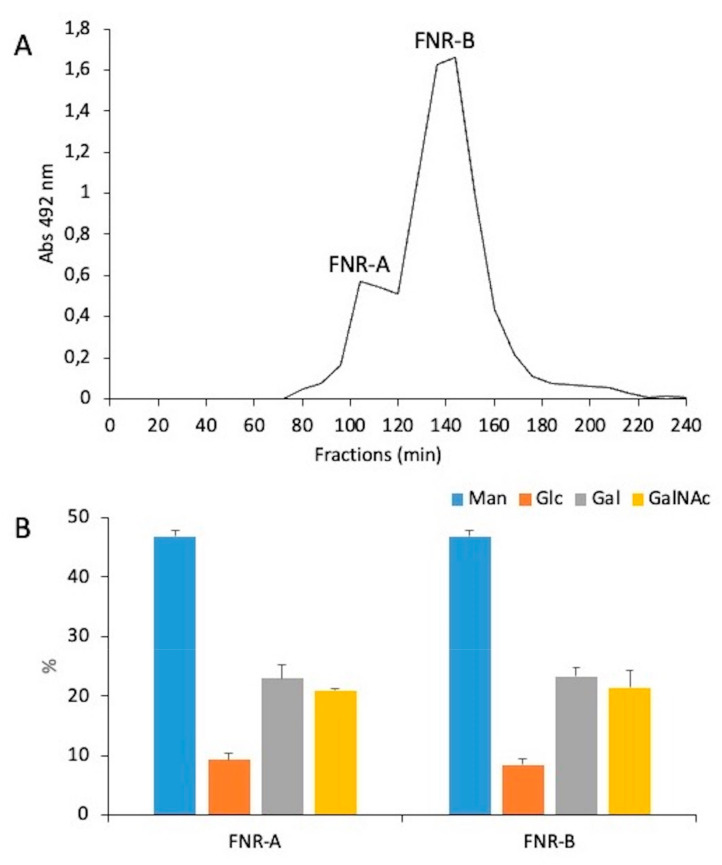
Analysis of the mannan polymer of the ASSN fraction extracted from the resting conidial cell wall. (**A**) Gel filtration chromatography on a Superdex S200 column of mannan polymer isolated from the ASSN fraction of *Δugm1* mutant conidia and unbound on Q-Hitrap column. Sugars were detected at 490 nm by a colorimetric phenol-sulfuric method. (**B**) Monosaccharide composition of both mannan fractions is expressed relative to the total sugar content and performed on three independent biological replicates.

**Figure 4 jof-09-00155-f004:**
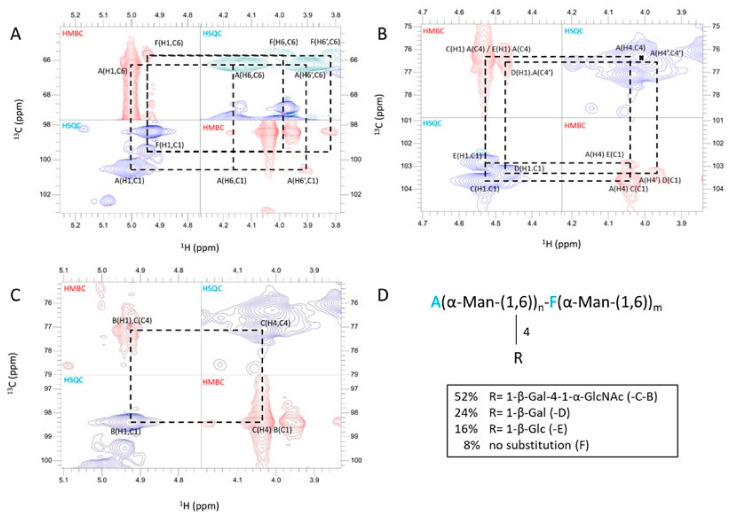
Superposition of ^1^H-^13^C HSQC (blue) and ^1^H-^13^C HMBC (red) NMR spectra of the FNR-B fraction. (**A**) Connectivity between α-(1,6)-mannose residues. (**B**) Connectivity of monosaccharide residues with 4-O-substituted α-(1,6)-mannose residues. (**C**) Connectivity of GlcNAc residues with 4-O-substituted galactose residues. (**D**) Scheme of the polysaccharide main structure. The relative abundance of each side chain (R) is indicated as a percentage.

**Figure 5 jof-09-00155-f005:**
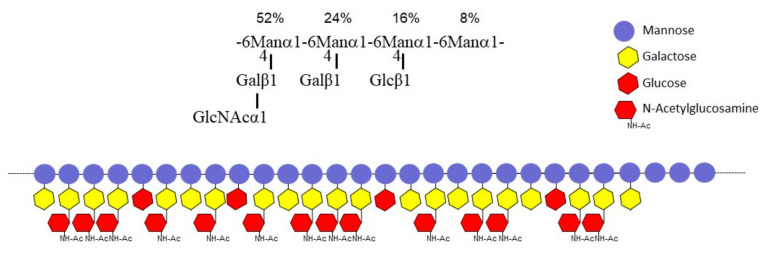
Proposed structure of the G3Man polysaccharide extracted from the *A. fumigatus* conidial cell wall. The backbone is composed of a linear α-(1,6)-mannan chain with short side chains. The three types of lateral chains and their relative abundance were established by NMR. Their distribution is unknown; however, NMR experiment showed the presence of substitution-free α-(1,6)-mannose sequence.

**Figure 6 jof-09-00155-f006:**
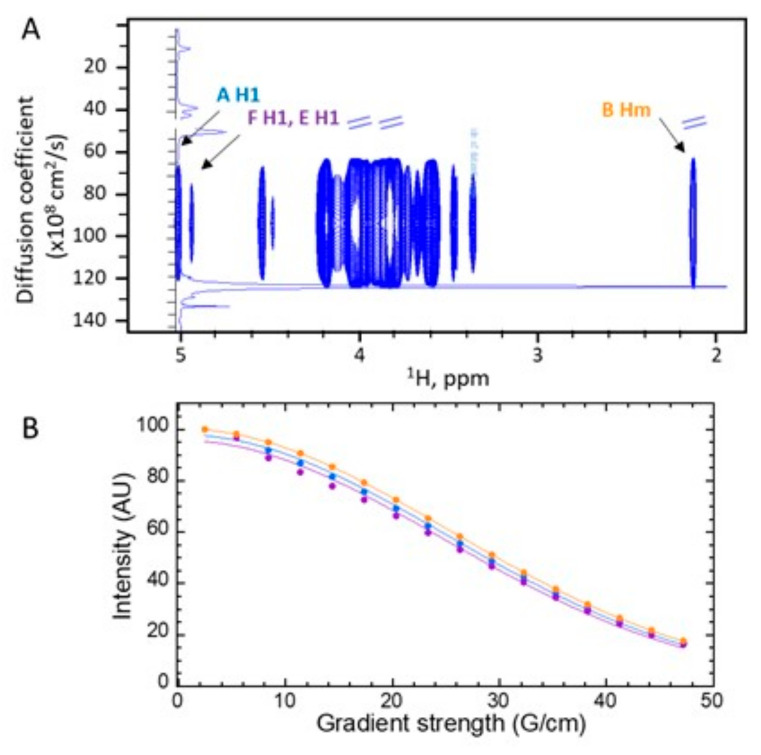
^1^H DOSY spectrum of the mannan polymer contained in the size exclusion chromatography FNR-B fraction obtained from the ASSN fraction (**A**). Experiments were recorded at 298°K. The center and width of the signals in the diffusion pseudo-dimension represent the mean and standard deviation of the diffusion coefficient at each resonance frequency, respectively. The intensity is proportional to the number of protons that contribute to the signal. All signals show the same mean and distribution (95 ± 31 × 10^−8^ cm^2^/s), indicating that the signals pertain to the same molecule. The 1D ^1^H spectrum is overlaid on the DOSY spectrum. (**B**) Selected diffusion curves (signal integral as a function of the gradient strength) that fit to single gaussian decays for the signals of the anomeric protons of 4-O-substituted ((**A**), 5 ppm, blue), unsubstituted (F) α-(1,6)-mannose (4.96 ppm, purple) that overlaps with the Glc (E) H1 signal, and of GlcNAc methyl (Hm) protons ((**B)**, 2.12 ppm, orange).

**Figure 7 jof-09-00155-f007:**
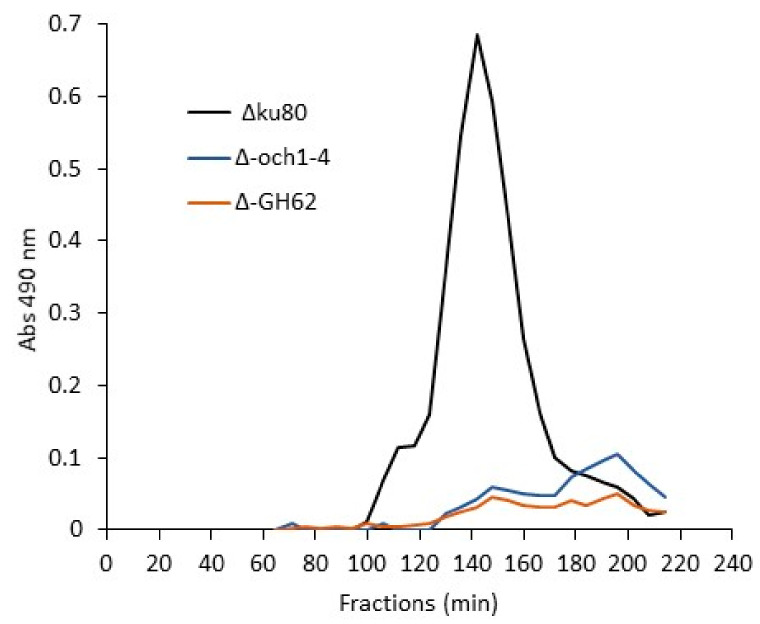
Gel filtration chromatography of the G3Man polymer isolated from the ASSN of *Δku80, Δoch1-4* and *ΔGH62* (Δmnn9/Δvan1/Δanp1 triple mutant) conidia on a Superdex S200 column. Sugars were detected by the colorimetric phenol-sulfuric method.

**Table 1 jof-09-00155-t001:** Methylation of the FNR-A and FNR-B fractions.

Methyl-Ethers ^1^	Linkages	FNR-A (%) ^2^	FNR-B (%) ^2^
2,3,4,6-Glc	t-Glc	9.7	9.6
2,3,4,6-Gal	t-Gal	8.0	8.9
2,3,6-Gal	4-Gal	22.8	17.5
2,3,4-Man	6-Man	5.8	7.1
2,3-Man	4,6-Man	47.0	50.6
3,4,6-GlcNAc	t-GlcNAc	6.7	6.1

^1^ methyl-ethers are identified and quantified by GC-MS as partially methylated and acetylated alditol. Numbers before sugar represent the position of methyl groups. ^2^ Percentages were calculated from peak areas on GC-MS.

**Table 2 jof-09-00155-t002:** ^1^H and ^13^C NMR resonance assignments and ^3^J_H1/H2_ coupling constants of the monosaccharide residues. Monosaccharides are identified by a letter code. Chemical shifts are expressed in ppm and coupling constants in Hz. Hm and Cm are the ^1^H and ^13^C chemical shifts of the N-acetyl methyl group, respectively.

Residue	H1, C1 ^3^J_H1/H2_	H2, C2	H3, C3	H4, C4	H5, C5	H6, H6’, C6	Hm, Cm
A -6,4)-Man-α(1-	5.00 100.6 1.8 Hz	4.17 70.0	4.00 70.4	4.01 76.3	4.00 69.5	3.90 4.14 66.2	
B GlcNAc-α(1-	4.93 98.4 ~2 Hz	3.95 54.1	3.84 70.6	3.58 69.9	4.18 72.2	3.86 3.86 60.3	2.12 22.0
C -4)-Gal-β(1-	4.53 103.7 8.9 Hz	3.61 70.94	3.78 72.5	4.04 77.1	3.79 72.5	3.66 3.66 70.8	
D Gal-β(1-	4.47 103.3 8.5 Hz	3.58 71.1	3.72 72.9	3.98 68.8	3.72 72.9	3.62 3.62 71.1	
E Glc-β(1-	4.54 102.8 9.2 Hz	3.36 73.3	3.57 76.0	3.46 69.8	3.56 76.4	3.78 3.98 61.0	
F -6)-Man-α(1-	4.94 99.5 ~2 Hz	4.03 70.1	3.87 71.0	3.77 66.7	3.88 70.9	3.83 3.97 65.7	

**Table 3 jof-09-00155-t003:** Polysaccharidome ^a^ comparison of the conidial and mycelial cell wall of *A. fumigatus*.

	Conidium	Mycelium
β-glucans ^b^	49	33
α-(1,3)-glucans	7	30
chitin	10	18
GAG	-	7
GM	15	11
G3Man	18	-

^a^ composition of conidia and mycelium is expressed in % of the total sugar. ^b^ β-glucans mainly consist of β-(1,3)-glucans; a minor amount of β-(1,3)(1,4)-glucan was detected but not quantified.

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
