# Peer review of "Conidium Specific Polysaccharides in Aspergillus fumigatus"

_jof, 2023, doi:10.3390/jof9020155_

Round 1

Reviewer 1 Report

This is an important work summarizing the conidial sugar cell wall composition using chemical and mutant analyses. It should be made more accessible to a non cell wall mycologist by various additions and clarifications.

Abstract- The abstract is not clear. Please be more specific. What is meant by “disparity in the organization of constitutive polysaccharides” and “a change in the solubility and repartition” etc. Please be more specific. Conclusions in abstract are missing. Please add.

Intro

L41 “the underneath” correct to “the underlying”

L56 “Later” to “latter”

Results

Was there any difference in conidial wall composition between those grown on malt agar and those on minimal medium? Please add?

L166-207. Were mycelial cell wall fractions analyzed and compared in parallel? Please add if so and where possible compare the numbers. Most readers would appreciate a comparison, perhaps in a Table. For example, L177, how much water soluble beta glucan was found in the mycelial cell wall?

Are the results shown in Fig. 1-3 that of a single experiment? How many independent samples in this experiment? Please add. Are the results shown here from growth on malt or AMM? Please add.

Fig 3B- why are there no error bars here?

Tables- need to be in same format and font. Where is Table 3?

Discussion

L339 what is meant by “The organizational the cell wall” and “relies mainly in the difficulty”

L348 and elsewhere why do you call it a “polysaccharidome analysis”? isn’t it a “polysaccharide analysis”?

L349- I cannot see Table 3 anywhere. Please add. It is critical to display mycelial vs. conidial polysaccharide findings

I would add a summary figure of mycelial vs. conidial cell wall structure incorporating the results previously known and highlighting those found here.

Author Response

Authors thank the reviewer for its comments and remarks that rise up the quality of the manuscript.

Point 1 : The abstract is not clear. Please be more specific. What is meant by disparity in the organization of constitutive polysaccharides and a change in the solubility and repartition etc. Conclusions in abstract are missing.

Response : The abstract has been rewritted to include more results. We include details on the characteristic of conidia cell wall, the analyses of cell wall mutants (GH-72, GT-32 and GT-62) and their role in cell wall organisation.

The abstract is now the following :

Earlier studies have shown that the outer layer of conidial and mycelial cell walls of Aspergillus fumigatus are different. In this work, we analyzed the polysaccharidome of the resting conidial cell wall and observed major differences with the mycelium cell wall. Mainly, the conidia cell wall was characterized by : (i) a smaller amount of α-(1,3)-glucan and chitin; (ii) a larger amount of β-(1,3)-glucan, which was divided into alkali-insoluble and water-soluble fractions, and (iii) the existence of a specific mannan with side chains containing galactopyranose, glucose and N-acetylglucosamine residues. Analysis of A. fumigatus cell wall gene mutants suggested that members of the fungal GH-72 transglycosylase family play crucial role in the conidia cell wall β-(1,3)-glucan organization and that α-(1,6)-mannosyltransferases of GT-32 and GT-62 families are essential to the polymerization of the conidium-associated cell wall mannan. This specific mannan and the well-known galactomannan follow two independent biosynthetic pathways.

Point 2 and Response : Spelling mistakes have been corrected

Point 3 : Was there any difference in conidial cell wall composition between those grown on malt agar and those on minimal medium?

Response :

Analysis of cell wall composition of parental strain and mutants has always be performed on conidia grown on malt media. The minimum medium/phytagel was used only for the purification of the G3Man polymer. Indeed, to prevent any contamination from Malt or agar, we switch to AMM/phytagel. To clarify, we added a sentence in the MM section (Fungal strains and growth media). Conidia grown on malt medium were used from the cell wall analysis and conidia grown on AMM/phytagel medium were used to purify and characterize the G3Man polymer.

Point 4 :

L166-207. Were mycelial cell wall fractions analyzed and compared in parallel? Please add if so and where possible compare the numbers. Most readers would appreciate a comparison, perhaps in a Table. For example, L177, how much water soluble beta glucan was found in the mycelial cell wall?

Response :

To clarify on the ASSN fraction in Mycelium, we add a new supplementary figure (Suppl. Fig.1) showing the comparison of cell wall fractions between mycelium and conidia. As indicated I the text, ASSN fraction is 2.5% of the total cell wall in mycelium and around 25% in conidia.

Point 5 : Are the results shown in Fig. 1-3 that of a single experiment? How many independent samples in this experiment? Please add. Are the results shown here from growth on malt or AMM?

Fig. 3B-Why are they no error bar here?

Response

Cell wall analyses in Figs 1-2 and supplementary Fig-2 have been performed on three independent biological replicates grown on malt medium. It is now indicated in legends.

For the panel B of Fig.3, it was the composition of the fraction corresponding to the panel A. We change the panel B to indicate the average composition from three independent purifications.

Point 6 : Tables need to be in same format and font. Where is table 3

Response

The table 3 is now included in the text. Sorry for this mistake. This table shows the comparative composition of cell wall polymer between conidia and mycelium and answer to point 4 and help the understanding for not cell wall specialists.

All tables are now in same format and Calibri police.

We hope that all these precisions and corrections respond to the reviewer 1 comments.

Reviewer 2 Report

There are only a few very specific corrections, in the spaces between words.

Line 22 there is an space before the ;

Line 48 there is an space after de suggested word

Line 35 there is an space after glucan of and another space after de by a

This work is very interesting because open door future research about the biosynthetic pathways of the mannan structures. Also is very important due to connect the diagnosis and prognostic about the Aspergillus and the galactomannan test, because the principal component that initiates the infection is the conidia and it has a different composition, and the immune response is different. 

Author Response

The authors thank the reviewer for his kind comment.

All suggested corrections have been made.

Reviewer 3 Report

The manuscript describes a very thorough examination of the Aspergillus fumigatus conidial inner cell wall.  It does an excellent job of describing a newly identified mannan structure found in the conidial wall as well as characterizing the presence of beta-1,3-glucan and mixed beta-1,3-/1,4-glucans in the cell wall.   There are two suggestions of a more general nature that I would like to make.   

1)  When a fungal cell wall is compromised, the fungus activates a cell wall integrity pathway leading to the synthesis of additional cell wall proteins and polysaccharides to compensate for the compromised cell wall.  This complicates how one might think about the changes in the conidial cell wall that the authors describe for the gel-1/gel-2 mutant.   The changes could be directly due to loss of the GH72 enzymes gel-1/gel-2 or the changes might be the mixed result of the loss of GH72 activity and the induction of the cell wall integrity pathway with the changes in cell wall proteins and polysaccharides that occur when the pathway is activated.  I would like to suggest that the authors briefly discuss the presence of the cell wall integrity pathway and indicate how it might affect the cell wall composition they see in the gel-1/gel-2 mutant.

2) They authors show that the och-1 and Mnn9/Van1/Anp1 mannosyltransferases are required for the synthesis of their newly identified G3Man polysaccharide, which they show has an alpha-1,6-backbone.  They correctly mention that these enzymes have been shown to be needed for the synthesis of N-linked mannans (with alpha-1,6-mannose backbones).  Their discussion doesn't seem to touch on some important ideas for how och-1 and Mnn9/Van1/Anp1 might be involved in G3Man synthesis.  Clearly they could be directly needed for the synthesis of the G3Man backbone, but the question of whether G3Man is produced as an N-linked modification that is subsequently released by a mannanase into the cell wall or is made without attachment to a N-linked oligosaccharide is not well discussed.  I think och-1 requires an N-linked oligosaccharide as a substrate.   I would think a more extensive discussion of how och-1 and the other mannosyltransferases are involved in G3Man synthesis ought to be included.  I think there is a very slight possibility that the involvement of och1 and Mnn9/Van1/Anp1 could be secondary - that the synthase(s) for G3Man might require extensive N-linked modification for folding and stability and that the loss of N-linked modifications might result in the G3Man synthase(s) being unable to correctly fold or to be rapidly degraded, resulting in the loss of G3Man.         

Author Response

Authors thank the reviewer for his kind comments on our manuscript.

1)  When a fungal cell wall is compromised, the fungus activates a cell wall integrity pathway leading to the synthesis of additional cell wall proteins and polysaccharides to compensate for the compromised cell wall.  This complicates how one might think about the changes in the conidial cell wall that the authors describe for the gel-1/gel-2 mutant.   The changes could be directly due to loss of the GH72 enzymes gel-1/gel-2 or the changes might be the mixed result of the loss of GH72 activity and the induction of the cell wall integrity pathway with the changes in cell wall proteins and polysaccharides that occur when the pathway is activated.  I would like to suggest that the authors briefly discuss the presence of the cell wall integrity pathway and indicate how it might affect the cell wall composition they see in the gel-1/gel-2 mutant.

Response: We agree with the suggestion. The increase of alpha-(1,3)-glucan in the cell wall composition of the GH-72 mutant is not due to the absence of gel1/2 activity but it is a compensatory salvage. We add the following sentence in the discussion (lines 448-450): “The strong increase of α-(1,3)-glucan content in conidia cell wall of Δgel1/Δgel2 mutants (Fig. 2) suggests a compensatory effect through the activation of the cell wall integrity pathway as has been previously observed in Aspergillus species [51,52].”

We included two new references.

2) They authors show that the och-1 and Mnn9/Van1/Anp1 mannosyltransferases are required for the synthesis of their newly identified G3Man polysaccharide, which they show has an alpha-1,6-backbone.  They correctly mention that these enzymes have been shown to be needed for the synthesis of N-linked mannans (with alpha-1,6-mannose backbones).  Their discussion doesn't seem to touch on some important ideas for how och-1 and Mnn9/Van1/Anp1 might be involved in G3Man synthesis.  Clearly they could be directly needed for the synthesis of the G3Man backbone, but the question of whether G3Man is produced as an N-linked modification that is subsequently released by a mannanase into the cell wall or is made without attachment to a N-linked oligosaccharide is not well discussed.  I think och-1 requires an N-linked oligosaccharide as a substrate.   I would think a more extensive discussion of how och-1 and the other mannosyltransferases are involved in G3Man synthesis ought to be included.  I think there is a very slight possibility that the involvement of och1 and Mnn9/Van1/Anp1 could be secondary - that the synthase(s) for G3Man might require extensive N-linked modification for folding and stability and that the loss of N-linked modifications might result in the G3Man synthase(s) being unable to correctly fold or to be rapidly degraded, resulting in the loss of G3Man.        

Response: We think that GT-32 and GT-62 α-1,6-mannosyltransferases are responsible to the polymerization of the α-1,6-mannan backbone of the G3Man polymer and consequently G3Man is part of N-glycan of cell wall glycoproteins. Unfortunately, we never succeed to extract the G3Man without destructive method. So, we did not show any covalent linkage between G3Man and protein.

About the putative secondary effect of GT-32 and GT-62, we do not believe this hypothesis. Although we can’t not totally exclude it, we observe that deletion of GT-32 and GT-62 members has no effect on conidiation and vegetative growth in A. fumigatus (Henry et al., 2016), suggesting that the absence of such activities is not required for other pathway. We have chosen not to mention this alternative hypothesis.

To respond to this suggestion, we have added the following sentences in the discussion (lines 380-387) :

Our data suggest that Ochp and GT-32 members are responsible to the polymerization of the α-(1,6)-mannan backbone of the G3Man and therefore suggest that G3Man is a part of N-glycan of cell wall glycoproteins. Unfortunately, we were unable to extract the cell wall G3Man without destructive method (1M NaOH, 65°C), preventing the identification of a G3Man-protein glycoconjugate, but suggesting that G3Man is tighly attached to the cell wall AI core in its native form.
